# Global and Chinese epidemiologic study of polycystic ovary syndrome in women of childbearing age, 1990–2021, and projections to 2035: Based on the Global Burden of Disease 2021 study

Junping Liu[1], Yan Cai[2], Jiaying Li[3], Xiaodan Zhang ![ORCID][4]*

**1** Department of Traditional Medicine, Seventh People's Hospital, Shanghai University of Traditional Chinese Medicine, Shanghai, China, **2** Department of Traditional Medicine, Seventh People's Hospital, Shanghai University of Traditional Chinese Medicine, Shanghai, China, **3** Department of Traditional Medicine, Seventh People's Hospital, Shanghai University of Traditional Chinese Medicine, Shanghai, China, **4** Department of Traditional Medicine, Seventh People's Hospital, Shanghai University of Traditional Chinese Medicine, Shanghai, China

\* 2513840348@qq.com

## Abstract

### Objective

The aim of this study was to synthesize and analyze the burden of disease of polycystic ovary syndrome (PCOS) in women of reproductive age globally and in China from 1990 to 2021.

### Methods

The study utilized data from the Global Burden of Disease 2021 database, which contains detailed epidemiologic information from 204 countries and territories. Incidence, prevalence and disability-adjusted life years (DALYs) of polycystic ovary syndrome (PCOS) were assessed. Bayesian age-period cohort modeling was applied to project trends in the burden of disease up to 2035.

### Results

Global trend: the incidence of polycystic ovary syndrome (PCOS) increased slightly from 58.84/100,000 in 1990 to 60.30/100,000 in 2021. there was a significant increase in prevalence of 29.66% and an increase in disability-adjusted life-years (DALYs) of 28.37%. Trends in China: Despite a slight decrease in prevalence (−1.96%), the prevalence of polycystic ovary syndrome (PCOS) and the number of DALYs in China increased substantially by 86.95% and 86.56%, respectively. Sociodemographic impact: Countries with higher sociodemographic indices tend to face a higher burden of PCOS. Future projections: BAPC model projections suggest that the prevalence of PCOS will continue to increase globally and in China through 2035.

**Data availability statement:** All relevant data are within the manuscript and its Supporting information files.

**Funding:** This work was supported by the Shanghai Municipal Healthcare Commission Healthcare Program (202340155); Shanghai Pudong New District Clinical TCM Characteristic Discipline Construction Grant (YC-2023-0611); Shanghai Pudong New District Health System Leading Talent Training Program (PWRl2023-10); Shanghai Municipal Health Commission Traditional Chinese Medicine Research Project (2024BJ018); Discipline Leader, Pudong New District Health Committee (PWRD2022-20). The funders had no role in study design, data collection and analysis, decision to publish, or preparation of the manuscript.

**Competing interests:** The authors have declared that no competing interests exist.

## Conclusion

These findings highlight the growing public health challenge posed by polycystic ovary syndrome and emphasize the need to strengthen early identification, health management, and lifestyle interventions, especially in areas with high SDI.

## Introduction

Polycystic ovary syndrome (PCOS) is a widespread and complex endocrine disease that affects 5–20% of women of childbearing age and has a profound impact on their reproductive, metabolic and mental health [1,2]. PCOS, characterized by hyperandrogenism, oligoanovulation, and polycystic ovarian morphology according to the 2023 international evidence-based guideline, is the leading cause of anovulatory infertility. The syndrome is associated with an increased risk of cardiovascular disease, endometrial cancer, and significant psychosocial impairment, alongside dermatological manifestations including hirsutism, acne, and alopecia [3,4].The syndrome also imposes a significant financial burden on the healthcare system. It is estimated that the annual healthcare-related economic burden due to polycystic ovary syndrome is as high as $4.3 billion in the United States alone [5].

In China, Chinese medicine categorizes PCOS as "irregular menstruation" and "infertility" based on its clinical manifestations. TCM believes that the onset of PCOS is related to the dysfunction of the liver, spleen, kidney and other internal organs [6]. In treatment, TCM emphasizes the harmonization of yin and yang, tonifying qi and blood, and relieving the liver and depression, with a view to restoring normal menstrual cycles and promoting ovulation [7]. Modern research has also confirmed [8] that TCM treatment can improve the endocrine status of PCOS patients, reduce the level of androgens and increase insulin sensitivity, which has a positive effect on relieving symptoms and enhancing fertility.

Despite its high incidence and impact, PCOS remains complex to study due to its heterogeneity and varying diagnostic criteria used worldwide [9]. The Rotterdam criteria, the National Institutes of Health criteria, and the Hyperandrogenic-Polycystic Ovary Syndrome Association criteria all provide different diagnostic methods for PCOS, leading to possible differences in estimated prevalence and understanding of the global burden of the disease [10].

The global burden of PCOS is increasingly recognized through global burden of disease studies. The GBD 2019 data comprehensively assessed the incidence, prevalence, and disability-adjusted life years of PCOS in 204 countries and territories between 1990 and 2019 [11].This study shows worrying increases in the prevalence, incidence and disability-adjusted life years of age-standardized rates due to PCOS worldwide, and there are also significant differences among different regions and countries.

China has a large population and diverse ethnic groups, which provides a unique background for studying PCOS. The rapid economic development and lifestyle changes in China have led to changes in the incidence of metabolic diseases

including PCOS [12]. However, comprehensive and nationwide studies on the epidemiology of PCOS in China, especially those using the latest and most reliable data sources, are still lacking.

This study aims to fill this gap with a comprehensive analysis of the burden of PCOS globally and in China from 1990 to 2021, with a special focus on China. By leveraging the latest data from the GBD 2021 study, this study will provide insights into epidemiological trends of PCOS globally and in China, the potential impact of socio-demographic changes on its prevalence and burden.

## Materials and methods

### Data sources

In this study, we used the Global Burden of Disease (GBD) 2021 database, which records detailed epidemiologic information on 371 different diseases and injuries in 204 countries and territories around the world from 1990 to 2021. To ensure global consistency of findings, this study followed the GBD standardized epidemiological analysis procedures, which harmonize data handling, analytical methods, and statistical frameworks. These datasets are publicly accessible and freely available through the Global Health Data Exchange website [13]. The associated analytical techniques and modeling framework have been detailed in earlier publications [14]. Specifically for this study, we retrieved epidemiologic indicators on PCOS, including its incidence, prevalence, and resulting disability-adjusted life years, from the database. The retrieval and export of data was done through the GHDx platform, the relevant link is: (http://ghdx.healthdata.org/gbd-results-tool).

### Important definitions

The socio-demographic index (SDI) is an important indicator of a society's level of economic development, covering multidimensional factors such as education, income level and fertility [15].The SDI takes values between 0 and 1, with larger values indicating a higher level of economic and social development in that society. Based on the SDI data from the GBD 2021 study, the world's 204 countries and territories were categorized into five different SDI levels: high, medium-high, medium, medium-low, and low. Uncertainty Interval (UI) is a statistical tool used in the GBD study, which is calculated through multiple sampling and correlation analyses to reflect the different methods and potential data deficiencies in international data collection and processing.The UI reveals the variability in global data collection and processing, as well as the reality that data quality is affected by a variety of factors, and is useful for evaluating the credibility of data across countries. The UI reveals the differences in global data collection and processing, as well as the fact that data quality is affected by a variety of factors. Confidence intervals (CI), on the other hand, are statistically derived to infer the range in which the overall parameter is likely to fall, and are calculated based on the sample data. In this study, we extracted data on incidence and prevalence from the GBD database and applied UI to label the uncertainty of these data in order to demonstrate more precisely the confidence of the findings.

### Statistical analysis

The global burden of disease for PCOS is measured by several different indicators, including incidence, prevalence and disability-adjusted life years (DALYs). To understand the trend in the prevalence of PCOS between 1990 and 2021, we calculated the annual percentage change (EAPC), which was derived from a linear regression model [16]. If both the EAPC value and the lower limit of its 95% confidence interval are greater than zero, this indicates an increasing age-standardized rate. If both the EAPC value and the upper limit of its 95% confidence interval are less than zero, this indicates that the ASR is decreasing [17]. If neither condition is met, then the ASR is considered stable. Spearman's rank correlation coefficient was used to assess the correlation between these rates of PCOS and SDI.

### Prediction

To project trends in the burden of disease for PCOS through 2035, we used a Bayesian age-period cohort (BAPC) model [18,19]. This model utilizes Integrated Nested Laplace Approximation for complete Bayesian inference, providing a robust framework for prediction [20]. All statistical analyses were done using the R package (version 4.2.1). Results were considered statistically significant if the two-sided p-value was less than 0.05.

### Ethics approval

Ethics approval was exempted by the Ethics Committee of Seventh People's Hospital Affiliated to Shanghai University of Traditional Chinese Medicine because the GBD 2021 study is a publicly available database and all data were anonymous.

## Results

### Global and Chinese disease burden of PCOS in women of reproductive age

A total of 204 countries worldwide were included in the study to assess the global and Chinese burden of polycystic ovary syndrome disease in women of reproductive age. The study showed that the global incidence and prevalence of PCOS in women of reproductive age showed an increasing trend from 1990–2021.

The results of the study showed that the global incidence of PCOS increased slightly from 58.84/100,000 in 1990 to 60.30/100,000 in 2021 (Table 1). At the same time, the prevalence rate increased significantly from 2602.62/100,000 in 1990 to 3374.68/100,000 in 2021, an increase of 29.66% (Table 2). In addition, disability-adjusted life years due to PCOS also increased from 23.03/100,000 to 29.56/100,000, a 28.37% increase (Table 3). At the national level, China's data showed a similar growth pattern to the global trend, but with differences in growth rates. The incidence of PCOS in China declined slightly over this time period to −1.96% (Table 1), which contrasts with the slightly increasing global trend. However, PCOS prevalence and DALYs in China showed a significant increase of 86.95% and 86.56% (Table 2), (Table 3).

### Relationship between the burden of P\COS and the level of socio-economic development

Global burden of disease data for PCOS show significant regional differences between 1990 and 2021 (Fig 1).

From 1990 to 2021, the incidence of PCOS in women aged 15–49 years in the Eastern European region showed an increasing and then decreasing trend, with a incidence of 10.46 in 1990, reaching a peak of 11.77 in 2004, and then gradually decreasing to 10.37 in 2021; whereas globally the incidence in the same group gradually increased from 58.84 in 1990 increased to 60.30 in 2021, showing a slight fluctuating upward trend (S1 Table). During the same period, the prevalence of PCOS among women aged 15–49 years in Southeast Asia increased significantly from 2917.33 to 5457.54 yearly, indicating a doubling of the prevalence among women in this region, whereas the prevalence in Central Europe increased from 355.03 to 436.25, showing a trend of change in the prevalence in different regions (S2 Table). During the same period, the rate of disability-adjusted life years due to PCOS among females aged 15–49 years in the Andean Latin America region increased consistently from 39.75 to 54.70 (S3 Table).

### Incidence, prevalence and health burden of PCOS in women of different ages

The incidence, prevalence, and health burden of PCOS varies by age and disability-adjusted life years (Fig 2).

In 2021, global data on PCOS in women show that women aged 15–19 years have the highest incidence of 347.88/100,000 and women aged 45–49 years have the lowest incidence of 1.42/100,000 (S4 Table). The prevalence rate increased from 2523.76/100,000 for 15–19 year olds to 3182.50/100,000 for 45–49 year olds.(S5 Table); the DALYs indicator showed that the burden of disease for 15–19 year old females was 22.66/100,000 compared to 26.99/100,000 for 45–49 year old females.(S6 Table). This indicates that the health impact is more pronounced in younger women and that the prevalence increases but the burden of disease decreases relatively with age.

**Table 1. Incidence of global, regional, and chinese polycystic ovary syndrome in women of childbearing age, 1990-2021.**

Rate per 100 000 (95% UI)

| Location | 1990 Incidence cases | 1990 Incidence rate | 2021 Incidence cases | 2021 Incidence rate | 1990-2021 Cases change | EAPCs |
|---|---|---|---|---|---|---|
| **Global** | 786953.55 (510645.82,1176402.42) | 58.84 (38.18,87.96) | 1175074.43 (766845.46,1740577.56) | 60.30 (39.35,89.31) | 2.47 (−0.72,5.74) | 0.04 (−0.03,0.11) |
| **SDI quintile** | | | | | | |
| High | 252320.51 (151229.29,408339.29) | 111.30 (66.71,180.12) | 290179.16 (178955.45,446166.17) | 119.34 (73.60,183.49) | 7.22 (−2.43,19.91) | −0.22 (−0.44,−0.00) |
| High middle | 134122.54 (90376.00,197333.79) | 48.29 (32.54,71.05) | 144652.29 (95541.39,214274.39) | 47.41 (31.31,70.22) | −1.82 (−6.38,2.21) | −0.05 (−0.22,0.11) |
| Middle | 254927.13 (170409.99,369154.66) | 57.02 (38.12,82.57) | 378023.68 (251370.92,556456.73) | 61.12 (40.64,89.97) | 7.19 (1.70,12.02) | 0.30 (0.17,0.42) |
| Low middle | 110608.95 (73761.90,160444.59) | 40.53 (27.03,58.79) | 250783.22 (165236.38,370032.93) | 49.54 (32.64,73.09) | 22.22 (17.03,28.41) | 0.78 (0.72,0.83) |
| Low | 34466.67 (23052.64,50119.32) | 30.86 (20.64,44.88) | 110648.76 (72852.93,162526.09) | 40.34 (26.56,59.25) | 30.71 (24.49,36.40) | 0.91 (0.89,0.93) |
| **GBD region** | | | | | | |
| East Asia | 139854.48 (93197.22,203540.88) | 41.95 (27.95,61.05) | 135522.57 (88658.21,200718.04) | 40.96 (26.79,60.66) | −2.37 (−10.84,4.10) | −0.07 (−0.51,0.37) |
| Southeast Asia | 100073.58 (62990.08,153602.98) | 83.26 (52.41,127.80) | 185095.97 (116472.89,285779.88) | 101.03 (63.58,155.99) | 21.34 (12.12,30.49) | 0.92 (0.83,1.01) |
| Oceania | 964.40 (621.75,1431.61) | 62.06 (40.01,92.13) | 2424.63 (1583.80,3724.37) | 69.86 (45.63,107.30) | 12.55 (2.62,21.17) | 0.29 (0.14,0.44) |
| Central Asia | 2835.37 (1853.32,4154.69) | 16.90 (11.05,24.76) | 3985.96 (2608.26,5761.86) | 16.43 (10.75,23.75) | −2.80 (−9.01,2.77) | −0.06 (−0.22,0.11) |
| Central Europe | 2254.45 (1492.96,3341.69) | 7.34 (4.86,10.88) | 1603.08 (1093.45,2252.70) | 6.22 (4.25,8.75) | −15.20 (−21.00,-8.05) | −0.98 (−1.22,−0.74) |
| Eastern Europe | 5784.61 (4100.71,8021.99) | 10.46 (7.42,14.51) | 5003.87 (3581.13,6884.84) | 10.37 (7.42,14.27) | −0.85 (−6.00,4.10) | −0.19 (−0.38,−0.00) |
| High-income Asia Pacific | 109060.13 (59963.71,179392.20) | 238.43 (131.10,392.20) | 75026.08 (42893.90,116401.50) | 197.24 (112.76,306.01) | −17.28 (−24.04,-3.75) | −0.45 (−0.61,−0.28) |
| Australasia | 9332.19 (5720.27,14991.53) | 173.87 (106.57,279.30) | 11586.60 (6953.46,18883.29) | 160.50 (96.32,261.57) | −7.69 (−21.59,7.57) | −0.07 (−0.14,−0.01) |
| Western Europe | 89458.01 (61148.91,135388.60) | 93.63 (64.00,141.70) | 84435.70 (56831.71,128362.03) | 90.62 (60.99,137.76) | −3.21 (−6.31,-0.10) | −0.13 (−0.23,−0.02) |
| Southern Latin America | 7239.54 (4479.11,11333.26) | 58.42 (36.14,91.45) | 12557.10 (7864.40,19722.69) | 72.04 (45.12,113.15) | 23.33 (12.70,33.58) | 0.73 (0.45,1.02) |
| Caribbean | 4343.64 (2875.95,6472.52) | 46.60 (30.85,69.44) | 5373.28 (3619.78,7952.13) | 44.67 (30.09,66.11) | −4.14 (−9.39,1.22) | 0.14 (0.02,0.25) |
| Andean Latin America | 9224.21 (6105.68,14872.61) | 97.26 (64.38,156.81) | 15444.48 (10574.16,23485.90) | 88.49 (60.59,134.57) | −9.01 (−17.43,0.54) | −0.27 (−0.44,−0.11) |
| High-income North America | 73188.51 (43925.25,122662.33) | 98.41 (59.07,164.94) | 131668.61 (78605.91,202795.48) | 156.71 (93.55,241.36) | 59.23 (32.22,103.54) | 0.09 (−0.50,0.68) |
| Central Latin America | 40487.70 (28479.21,57586.24) | 96.60 (67.95,137.39) | 51387.51 (36692.32,72690.46) | 75.35 (53.81,106.59) | −21.99 (−24.48,-19.21) | −1.08 (−1.21,−0.95) |
| Tropical Latin America | 10701.13 (7355.01,15713.20) | 26.83 (18.44,39.39) | 12231.82 (8521.07,17488.87) | 20.18 (14.06,28.85) | −24.77 (−28.60,-20.43) | −1.38 (−1.61,−1.16) |
| North Africa and Middle East | 58625.68 (37961.81,89887.04) | 75.04 (48.59,115.06) | 108447.54 (71031.18,166927.54) | 68.06 (44.58,104.76) | −9.31 (−14.23,-5.52) | −0.42 (−0.57,−0.28) |
| South Asia | 85567.03 (59219.10,120515.57) | 33.57 (23.23,47.28) | 220589.13 (147780.96,314345.74) | 44.64 (29.91,63.62) | 32.98 (24.24,41.81) | 1.20 (1.09,1.31) |

*(Continued)*

**Table 1.** (Continued)

| Rate per 100 000 (95% UI) | | | | | | |
|---|---|---|---|---|---|---|
| Location | 1990 Incidence cases | 1990 Incidence rate | 2021 Incidence cases | 2021 Incidence rate | 1990-2021 Cases change | EAPCs |
| Central Sub-Saharan Africa | 3385.05 (2195.96,5004.05) | 27.38 (17.76,40.48) | 12222.81 (7924.44,18382.31) | 37.43 (24.27,56.30) | 36.70 (25.08,49.45) | 0.94 (0.80,1.08) |
| Southern Sub-Saharan Africa | 6290.23 (4122.32,9301.36) | 47.32 (31.01,69.98) | 9499.52 (6207.65,14037.03) | 43.75 (28.59,64.65) | −7.55 (−11.41,-3.84) | −0.36 (−0.54,-0.18) |
| Eastern Sub-Saharan Africa | 14331.50 (9430.42,21280.25) | 33.21 (21.85,49.32) | 41692.61 (27207.93,62169.02) | 38.93 (25.40,58.05) | 17.21 (12.62,22.19) | 0.51 (0.48,0.54) |
| Western Sub-Saharan Africa | 13952.09 (9257.16,20451.38) | 32.00 (21.23,46.90) | 49275.56 (32268.83,72559.81) | 41.10 (26.92,60.52) | 28.45 (21.91,34.68) | 0.55 (0.43,0.67) |
| Country | | | | | | |
| China | 134474.66 (89646.06,194982.78) | 41.74 (27.82,60.51) | 130403.31 (85354.87,193461.44) | 40.92 (26.78,60.70) | −1.96 (−10.71,4.73) | −0.06 (−0.52,0.40) |

## Global and Chinese projections of the disease burden of polycystic ovary syndrome in women of childbearing age

The projections of the BAPC model showed global and Chinese projection maps that the incidence and prevalence of polycystic ovary syndrome in women of reproductive age showed a significant increasing trend (S1–S4 Figs)

At the global level, especially in the age group of 15–19 years, the incidence rate increased yearly from 273.99/100,000 in 1990 to 462.69/100,000 in 2035 (S7 Table). The overall age-standardized incidence rate (ASIR) also increased from 52.10/100,000 to 84.74/100,000 (S8 Table), and in terms of prevalence, the projections found that: the prevalence rate in the age group of 15–19 years increased from about 1,925 cases per 100,000 to about 3,954 cases per 100,000 (S9 Table), the overall age-standardized prevalence rate (ASPR), the overall age-standardized prevalence rate increased from 2627.92 per 100,000 population to approximately 3812.48 per 100,000 population (S10 Table).

For China, in the age group 15–19 years, the prevalence rate increased from about 174 per 100,000 in 1990 to about 513 per 100,000 in 2035. The overall age-standardized incidence rate, increased from 36.88 per 100,000 to 95.67 per 100,000. In terms of prevalence, the projections found that the prevalence in the 15–19 age group is projected to increase from about 1,201 per 100,000 in 1990 to about 3,263 per 100,000 in 2035, and that the overall age-standardized prevalence rate is projected to increase significantly, from about 1,600 per 100,000 to about 4,000 per 100,000.

## Discussion

This study reveals that the incidence, prevalence, and disability-adjusted life years of PCOS have shown a significant increase globally from 1990 to 2021.

From 1990 to 2021, the global prevalence of PCOS increases slightly from 58.84/100,000 to 60.30/100,000, which is not a significant increase, but shows the continued prevalence of the disease globally. More significantly, the global prevalence of PCOS increased by 29.66% from 2602.62/100,000 to 3374.68/100,000, indicating that more and more women are being diagnosed with PCOS.In addition, there was a significant increase in the global burden of disease due to PCOS, with an increase of 28.37%. This trend is not only related to advances in diagnostic techniques worldwide, especially the use of imaging and biomarkers that have increased the diagnostic rate, but may also be closely related to the global

**Table 2. Global, regional and Chinese Prevalence of polycystic ovary syndrome among women of reproductive age (15-49 years), 1990-2021.**

**Rate per 100 000 (95% UI)**

| Location | 1990 Prevalence cases | 1990 Prevalence rate | 2021 Prevalence cases | 2021 Prevalence rate | 1990-2021 Cases change | EAPCs |
|---|---|---|---|---|---|---|
| **Global** | 34806508.02 (24932017.17,47919343.47) | 2602.62 (1864.27,3583.12) | 65767552.90 (46674562.58,90615556.30) | 3374.68 (2394.97,4649.68) | 29.66 (26.50,32.88) | 0.77 (0.74,0.81) |
| **SDI quintile** | | | | | | |
| High | 13171901.48 (9563341.91,18423100.94) | 5810.11 (4218.38,8126.40) | 16702141.31 (12277515.61,22711492.89) | 6868.85 (5049.20,9340.23) | 18.22 (12.52,26.28) | 0.09 (−0.08,0.27) |
| High middle | 6686758.24 (4720527.28,9252669.06) | 2407.44 (1699.54,3331.25) | 10547247.35 (7438447.28,14744927.97) | 3456.66 (2437.81,4832.38) | 43.58 (38.68,48.99) | 1.21 (1.16,1.25) |
| Middle | 10000306.17 (7026320.37,13878510.85) | 2236.89 (1571.66,3104.37) | 23243045.57 (16466866.83,32298675.97) | 3758.17 (2662.53,5222.38) | 68.01 (60.95,75.58) | 1.73 (1.68,1.77) |
| Low middle | 3864816.67 (2713568.85,5419512.79) | 1416.12 (994.29,1985.78) | 11502693.47 (8006074.18,16189725.04) | 2272.04 (1581.38,3197.84) | 60.44 (54.33,67.79) | 1.66 (1.61,1.70) |
| Low | 1059359.81 (740325.66,1514699.78) | 948.52 (662.87,1356.22) | 3726640.18 (2602170.53,5288064.00) | 1358.55 (948.62,1927.76) | 43.23 (37.56,50.46) | 1.24 (1.21,1.27) |
| **GBD region** | | | | | | |
| East Asia | 5386011.11 (3772179.96,7588876.46) | 1615.56 (1131.48,2276.32) | 9873929.85 (6960735.04,14017691.60) | 2983.92 (2103.55,4236.17) | 84.70 (72.89,96.32) | 2.06 (1.91,2.21) |
| Southeast Asia | 3506301.33 (2478196.23,4942003.17) | 2917.33 (2061.92,4111.88) | 9998314.58 (7055377.73,14098619.30) | 5457.54 (3851.15,7695.68) | 87.07 (74.63,100.38) | 2.31 (2.21,2.42) |
| Oceania | 37968.98 (26135.84,53568.50) | 2443.52 (1681.99,3447.43) | 117784.03 (81921.96,168230.89) | 3393.43 (2360.22,4846.83) | 38.87 (24.96,50.74) | 0.82 (0.65,1.00) |
| Central Asia | 112000.65 (75284.74,163264.24) | 667.59 (448.74,973.15) | 225487.58 (155209.26,316466.91) | 929.26 (639.64,1304.20) | 39.20 (28.55,48.76) | 1.18 (1.09,1.27) |
| Central Europe | 109037.60 (72300.72,162995.52) | 355.03 (235.42,530.72) | 112349.74 (76720.79,158461.04) | 436.25 (297.91,615.30) | 22.88 (9.39,37.37) | 0.65 (0.60,0.70) |
| Eastern Europe | 221989.38 (150603.77,321107.91) | 401.47 (272.37,580.73) | 251010.45 (175471.32,363218.05) | 520.31 (363.73,752.90) | 29.60 (23.10,35.83) | 1.04 (0.96,1.12) |
| High-income Asia Pacific | 4201471.33 (3030276.01,5851092.04) | 9185.53 (6624.98,12792.03) | 3894790.05 (2751824.10,5438023.31) | 10239.02 (7234.28,14296.03) | 11.47 (5.39,17.81) | 0.29 (0.24,0.33) |
| Australasia | 425029.09 (313038.72,559106.46) | 7918.64 (5832.17,10416.61) | 665141.92 (473655.68,925078.75) | 9213.66 (6561.16,12814.35) | 16.35 (2.05,33.84) | 0.27 (0.18,0.36) |
| Western Europe | 6457416.70 (4546423.77,8972735.57) | 6758.26 (4758.24,9390.76) | 7005904.34 (4935408.74,9775582.93) | 7518.71 (5296.66,10491.12) | 11.25 (7.19,15.40) | 0.20 (0.13,0.27) |
| Southern Latin America | 282120.49 (195050.99,409782.15) | 2276.47 (1573.89,3306.58) | 637611.34 (448562.29,915300.74) | 3658.16 (2573.53,5251.35) | 60.69 (48.40,75.76) | 1.49 (1.27,1.71) |
| Caribbean | 210459.69 (141599.31,302721.59) | 2257.80 (1519.07,3247.57) | 339901.77 (230827.00,490228.54) | 2825.76 (1918.97,4075.49) | 25.16 (18.73,31.86) | 0.75 (0.69,0.81) |
| Andean Latin America | 433322.94 (298407.98,603046.70) | 4568.87 (3146.35,6358.40) | 1105174.82 (756782.53,1547625.55) | 6332.51 (4336.27,8867.70) | 38.60 (26.43,51.70) | 1.10 (1.02,1.18) |
| High-income North America | 4287619.33 (3024720.01,6002205.00) | 5765.46 (4067.27,8071.03) | 6071379.68 (4532819.51,7994586.92) | 7225.93 (5394.79,9514.86) | 25.33 (10.46,48.60) | −0.53 (−1.04,-0.02) |
| Central Latin America | 2128215.00 (1460719.97,2956887.64) | 5077.66 (3485.10,7054.77) | 3806924.76 (2645028.80,5321794.89) | 5582.44 (3878.65,7803.84) | 9.94 (5.54,14.51) | −0.11 (−0.28,0.07) |
| Tropical Latin America | 418592.17 (283411.33,604423.15) | 1049.34 (710.46,1515.19) | 694299.33 (471531.34,984784.26) | 1145.47 (777.94,1624.71) | 9.16 (3.63,15.07) | −0.16 (−0.33,0.02) |
| North Africa and Middle East | 2314896.48 (1605499.08,3288153.70) | 2963.07 (2055.04,4208.84) | 6335262.73 (4447535.74,8950144.66) | 3975.80 (2791.13,5616.82) | 34.18 (27.55,41.38) | 1.11 (1.03,1.18) |
| South Asia | 3111983.01 (2216403.40,4330484.21) | 1220.90 (869.54,1698.95) | 10749370.06 (7599229.68,15053407.14) | 2175.38 (1537.88,3046.40) | 78.18 (67.43,92.00) | 2.16 (2.03,2.29) |

*(Continued)*

**Table 2.** (Continued)

| Rate per 100 000 (95% UI) | | | | | | |
|---|---|---|---|---|---|---|
| Location | 1990 Prevalence cases | 1990 Prevalence rate | 2021 Prevalence cases | 2021 Prevalence rate | 1990-2021 Cases change | EAPCs |
| Central Sub-Saharan Africa | 104326.33 (72179.09,151825.20) | 843.96 (583.90,1228.21) | 418511.19 (289388.67,604023.75) | 1281.70 (886.26,1849.84) | 51.87 (39.03,66.56) | 1.30 (1.15,1.44) |
| Southern Sub-Saharan Africa | 220249.05 (151839.42,316995.41) | 1657.00 (1142.33,2384.85) | 456795.93 (312538.88,644677.07) | 2103.88 (1439.47,2969.21) | 26.97 (20.93,33.31) | 0.83 (0.76,0.91) |
| Eastern Sub-Saharan Africa | 422757.12 (295394.67,609315.49) | 979.72 (684.57,1412.07) | 1362794.88 (957526.09,1946440.96) | 1272.44 (894.04,1817.39) | 29.88 (25.34,35.70) | 0.87 (0.84,0.91) |
| Western Sub-Saharan Africa | 414740.24 (289668.34,598696.04) | 951.13 (664.30,1373.00) | 1644813.85 (1150806.92,2346955.65) | 1371.94 (959.89,1957.59) | 44.24 (37.71,52.51) | 0.93 (0.74,1.12) |
| **Country** | | | | | | |
| China | 5127458.64 (3585815.74,7237269.41) | 1591.34 (1112.88,2246.13) | 9481520.05 (6677109.78,13483458.72) | 2975.09 (2095.13,4230.80) | 86.95 (74.48,99.02) | 2.10 (1.94,2.26) |

increase in health awareness and the focus on women's health issues [21]. However, the diagnostic criteria for PCOS vary according to geographic location, which may also be an important factor in the variation of epidemiologic data. Nonetheless, the global increase in the prevalence and disease burden of PCOS reflects the transformation of the disease from a purely reproductive health problem to a multifaceted public health challenge involving metabolic and psychological aspects.

China, as the most populous country in the world, is unique in its PCOS epidemiologic trends. Compared with global data, the incidence of PCOS in China decreased slightly (−1.96%) between 1990 and 2021, a phenomenon that may be related to several factors.

First, China has made remarkable progress in its ability to diagnose and recognize PCOS at an early stage in recent years. The development of medical technology, especially the popularization of ultrasound and hormone testing, has led to an increase in the rate of early diagnosis of PCOS, which to some extent may have contributed to the short-term decline in incidence.

Secondly, women's health awareness has been improving in China, and more and more women of childbearing age have begun to pay attention to their reproductive health and accept relevant examinations and diagnosis, thus affecting the morbidity statistics [22]. However, despite the decrease in the incidence of polycystic ovary syndrome (PCOS) in China, its prevalence and disease burden have increased significantly, by 86.95% and 86.56%, respectively. This increase is not only due to the increase in prevalence, but may also be related to environmental endocrine disruptors. In recent years, the potential impact of environmental endocrine-disrupting chemicals (EDCs) on polycystic ovary syndrome (PCOS) has received increasing attention [23]. EDCs are a class of exogenous chemicals widely found in plastics, pesticides, cosmetics, and industrial pollutants [24,25]. synthesis, metabolism, or signaling pathways, leading to abnormal reproductive and metabolic functions [26,27]. Existing studies have revealed multiple associations between EDCs exposure and the pathological mechanisms of PCOS: Available studies have shown that BPA exposure induces hyperandrogenemia and ovarian polycystic changes, and the mechanism involves aberrant activation of estrogen receptor β and dysregulation of the insulin signaling pathway [28,29].The urinary concentrations of phthalate metabolites were significantly higher in PCOS patients than in healthy populations, and were associated with insulin resistance, oxidative stress levels were positively correlated [30,31]; and the risk of ovarian cysts and the levels of inflammatory factors were significantly elevated in a group of women chronically exposed to industrial pollutants such as PCBs [32]. The continuous

**Table 3. Global, regional and Chinese DALYs of polycystic ovary syndrome among women of reproductive age (15-49 years), 1990-2021.**

**Rate per 100 000 (95% UI)**

| Location | 1990 DALY cases | 1990 DALY rate | 2021 DALY cases | 2021 DALY rate | 1990-2021 Cases change | EAPCs |
|---|---|---|---|---|---|---|
| **Global** | 307944.33 (136789.37,642228.92) | 23.03 (10.23,48.02) | 576045.19 (258039.94,1201744.64) | 29.56 (13.24,61.66) | 28.37 (24.90,31.70) | 0.74 (0.70,0.77) |
| **SDI quintile** | | | | | | |
| High | 116846.14 (52917.04,244334.66) | 51.54 (23.34,107.78) | 146877.29 (66875.12,300152.23) | 60.40 (27.50,123.44) | 17.20 (11.55,24.64) | 0.07 (−0.10,0.24) |
| High middle | 58694.21 (26219.70,122445.85) | 21.13 (9.44,44.08) | 91761.85 (40698.94,193892.03) | 30.07 (13.34,63.54) | 42.31 (36.55,47.62) | 1.18 (1.13,1.22) |
| Middle | 88354.00 (38869.59,186167.23) | 19.76 (8.69,41.64) | 203157.98 (90251.27,427042.75) | 32.85 (14.59,69.05) | 66.21 (58.90,73.88) | 1.69 (1.64,1.74) |
| Low middle | 34540.64 (14727.38,72964.80) | 12.66 (5.40,26.74) | 101239.02 (44101.84,213274.85) | 20.00 (8.71,42.13) | 58.00 (51.68,65.88) | 1.59 (1.55,1.63) |
| Low | 9301.24 (3960.42,19642.62) | 8.33 (3.55,17.59) | 32606.16 (13978.51,68975.40) | 11.89 (5.10,25.14) | 42.73 (36.55,50.19) | 1.21 (1.19,1.23) |
| **GBD region** | | | | | | |
| East Asia | 46319.95 (19893.87,96015.80) | 13.89 (5.97,28.80) | 84710.61 (37067.28,175598.65) | 25.60 (11.20,53.07) | 84.25 (71.63,96.05) | 2.05 (1.89,2.20) |
| Southeast Asia | 31336.72 (13686.84,63676.73) | 26.07 (11.39,52.98) | 88125.34 (39050.32,181684.80) | 48.10 (21.32,99.17) | 84.49 (71.91,98.23) | 2.25 (2.16,2.35) |
| Oceania | 332.77 (151.06,699.05) | 21.42 (9.72,44.99) | 1025.79 (444.35,2149.25) | 29.55 (12.80,61.92) | 38.00 (23.46,51.38) | 0.81 (0.63,0.98) |
| Central Asia | 990.38 (412.32,2120.88) | 5.90 (2.46,12.64) | 1972.54 (832.18,4258.55) | 8.13 (3.43,17.55) | 37.71 (27.30,48.97) | 1.15 (1.06,1.24) |
| Central Europe | 953.82 (399.43,1985.22) | 3.11 (1.30,6.46) | 974.40 (416.82,2047.96) | 3.78 (1.62,7.95) | 21.83 (8.05,38.11) | 0.63 (0.58,0.68) |
| Eastern Europe | 1966.87 (813.56,4144.12) | 3.56 (1.47,7.49) | 2198.96 (911.35,4610.16) | 4.56 (1.89,9.56) | 28.14 (21.08,36.08) | 1.00 (0.93,1.08) |
| High-income Asia Pacific | 36550.71 (16146.25,74038.30) | 79.91 (35.30,161.87) | 33722.68 (15178.09,68443.98) | 88.65 (39.90,179.93) | 10.94 (4.98,16.90) | 0.26 (0.22,0.31) |
| Australasia | 3709.28 (1688.45,7703.30) | 69.11 (31.46,143.52) | 5791.78 (2601.43,12019.81) | 80.23 (36.04,166.50) | 16.09 (1.80,32.41) | 0.27 (0.19,0.36) |
| Western Europe | 57716.93 (26244.29,120381.22) | 60.41 (27.47,125.99) | 62191.21 (28169.85,129165.60) | 66.74 (30.23,138.62) | 10.49 (5.98,14.56) | 0.18 (0.12,0.25) |
| Southern Latin America | 2505.98 (1122.04,5143.54) | 20.22 (9.05,41.50) | 5637.26 (2449.30,11708.59) | 32.34 (14.05,67.18) | 59.95 (46.25,75.67) | 1.48 (1.26,1.69) |
| Caribbean | 1876.57 (814.22,3907.37) | 20.13 (8.73,41.92) | 2981.76 (1294.64,6293.24) | 24.79 (10.76,52.32) | 23.13 (16.12,30.04) | 0.72 (0.66,0.78) |
| Andean Latin America | 3770.13 (1682.83,8161.75) | 39.75 (17.74,86.06) | 9546.48 (4191.75,19998.62) | 54.70 (24.02,114.59) | 37.61 (25.91,51.86) | 1.07 (1.00,1.15) |
| High-income North America | 38418.33 (16995.16,79942.88) | 51.66 (22.85,107.50) | 53692.33 (24466.83,108623.67) | 63.90 (29.12,129.28) | 23.70 (9.16,46.11) | −0.55 (−1.05,-0.04) |
| Central Latin America | 18714.33 (8296.51,39323.71) | 44.65 (19.79,93.82) | 33029.92 (14361.16,69176.90) | 48.43 (21.06,101.44) | 8.48 (3.67,13.42) | −0.13 (−0.30,0.03) |
| Tropical Latin America | 3751.88 (1603.17,7919.17) | 9.41 (4.02,19.85) | 6114.31 (2607.38,12901.94) | 10.09 (4.30,21.29) | 7.25 (1.97,13.05) | −0.20 (−0.37,-0.02) |
| North Africa and Middle East | 21107.04 (9225.35,44526.59) | 27.02 (11.81,56.99) | 56219.38 (25110.17,119097.30) | 35.28 (15.76,74.74) | 30.59 (22.86,37.67) | 1.02 (0.94,1.11) |
| South Asia | 27780.07 (12092.00,58787.35) | 10.90 (4.74,23.06) | 94327.65 (40874.85,197625.56) | 19.09 (8.27,39.99) | 75.15 (64.33,88.05) | 2.09 (1.97,2.21) |

*(Continued)*

**Table 3.** (Continued)

| Rate per 100 000 (95% UI) | | | | | | |
|---|---|---|---|---|---|---|
| Location | 1990 DALY cases | 1990 DALY rate | 2021 DALY cases | 2021 DALY rate | 1990-2021 Cases change | EAPCs |
| Central Sub-Saharan Africa | 905.43 (384.30,1873.21) | 7.32 (3.11,15.15) | 3640.90 (1570.98,7478.96) | 11.15 (4.81,22.90) | 52.23 (36.90,71.00) | 1.28 (1.12,1.45) |
| Southern Sub-Saharan Africa | 1939.55 (834.25,4237.24) | 14.59 (6.28,31.88) | 3956.56 (1694.51,8337.90) | 18.22 (7.80,38.40) | 24.88 (18.40,31.87) | 0.78 (0.70,0.86) |
| Eastern Sub-Saharan Africa | 3685.69 (1543.39,7733.23) | 8.54 (3.58,17.92) | 11841.69 (5066.96,25039.64) | 11.06 (4.73,23.38) | 29.45 (23.74,35.99) | 0.85 (0.82,0.88) |
| Western Sub-Saharan Africa | 3611.87 (1530.18,7577.81) | 8.28 (3.51,17.38) | 14343.66 (6144.18,30582.21) | 11.96 (5.12,25.51) | 44.44 (37.42,54.36) | 0.92 (0.73,1.11) |
| Country | | | | | | |
| China | 44078.48 (18966.74,91450.10) | 13.68 (5.89,28.38) | 81337.60 (35559.16,168662.71) | 25.52 (11.16,52.92) | 86.56 (73.26,98.68) | 2.08 (1.92,2.25) |

increase in the use of EDCs in China during the rapid industrialization process may partially explain the increasing trend in the prevalence of PCOS from an environmental exposure perspective [33,34].

With the rapid socio-economic development in China, especially urbanization and westernization of lifestyle [35], changes in dietary structure and rising body mass index have become important drivers of PCOS prevalence [36]. Therefore, the prevalence of polycystic ovary syndrome in China better reflects the complexity of rapid modernization and lifestyle changes.

According to the data in this study, countries with higher socioeconomic indices usually face higher prevalence and disease burden of PCOS [21]. This phenomenon is particularly prominent in China, especially in large cities and areas with rapid economic development. With China's rapid economic growth and accelerated urbanization, there have been significant changes in traditional dietary habits, with an increased intake of Western-style diets and high-sugar and high-fat foods, which have significantly increased the prevalence of metabolic syndrome, and consequently, the prevalence of PCOS [37]. Higher SDI levels usually imply better healthcare infrastructure and higher diagnostic rates, which has led to an increase in the diagnosis of PCOS in high-income countries and regions, reflecting the increased burden of the disease. However, in low-income or low-SDI countries, the disease burden of PCOS, although increasing more slowly due to deficiencies in health management and inadequate healthcare resources, is often overlooked and not recognized or intervened in a timely manner, which may lead to a potential threat of the disease to the future public health system of these countries. Therefore, the increase in SDI may be a driver of the increasing burden of PCOS, and it also provides a new direction for future PCOS prevention and treatment efforts, i.e., strengthening health management in high SDI countries and supporting medical resources in low SDI countries.

Projections of future PCOS disease burden through Bayesian age-period-cohort (BAPC) modeling show that the prevalence of PCOS globally and in China is projected to continue to grow through 2035. Further economic development and lifestyle changes worldwide are likely to exacerbate the prevalence of PCOS in the coming decades. Especially in emerging market economies, the prevalence of PCOS and the associated burden of disease will likely increase even more dramatically as lifestyles become further westernized. In addition, the trend toward later marriage and childbearing may lead to more women developing PCOS-related reproductive health problems as women's reproductive age is delayed. Therefore, future public health strategies need to identify high-risk groups in advance and strengthen early screening and intervention, especially preventive measures for metabolic diseases. At the same time, the development of science and technology and the sharing of cross-national health data will help to predict the accuracy of trends and provide more precise data to support global public health policymaking.

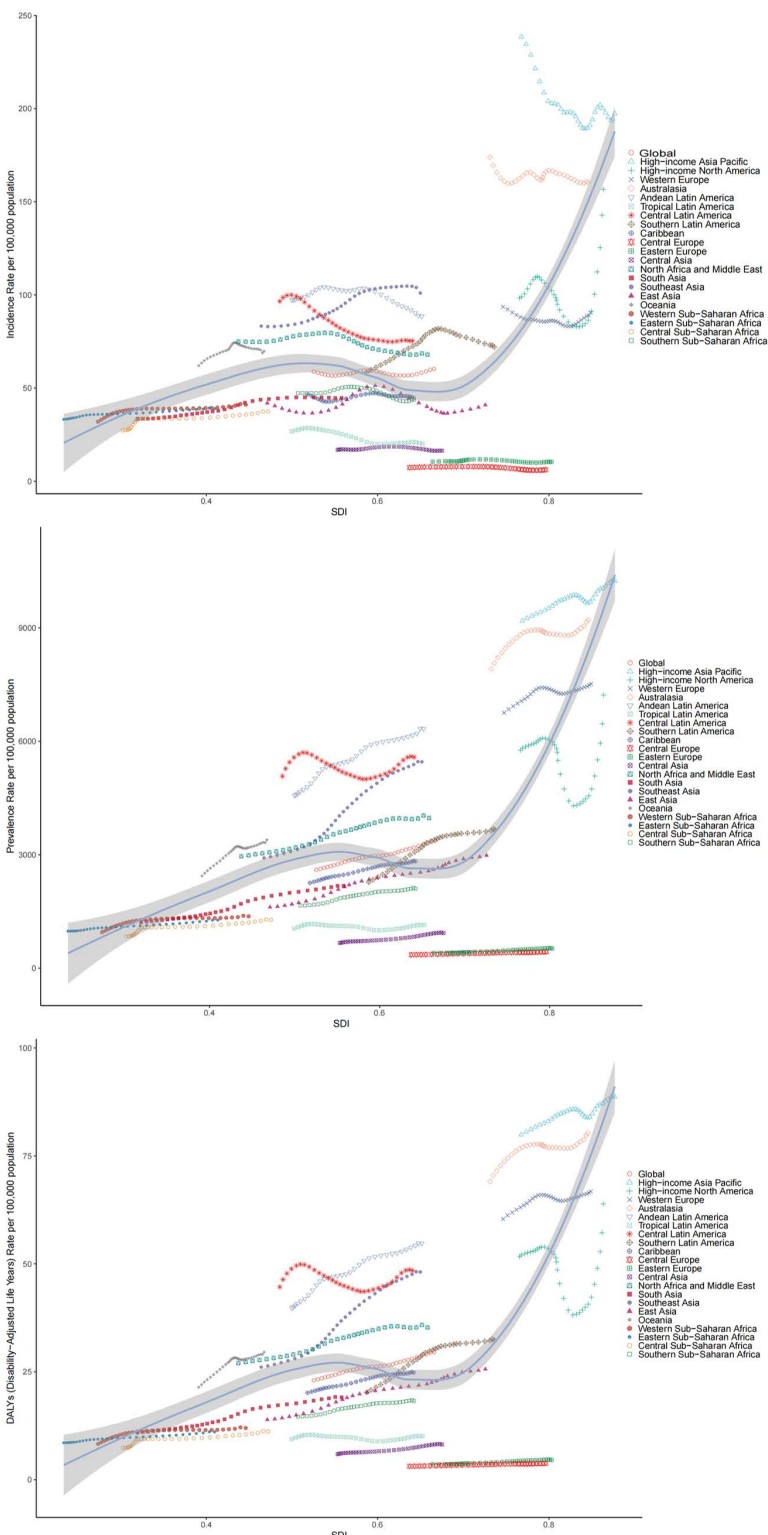

**Fig 1. Global comparative analysis of incidence, prevalence, and DALYs rates of polycystic ovarian syndrome among women aged 15-49 years across different regions and SDI levels.**

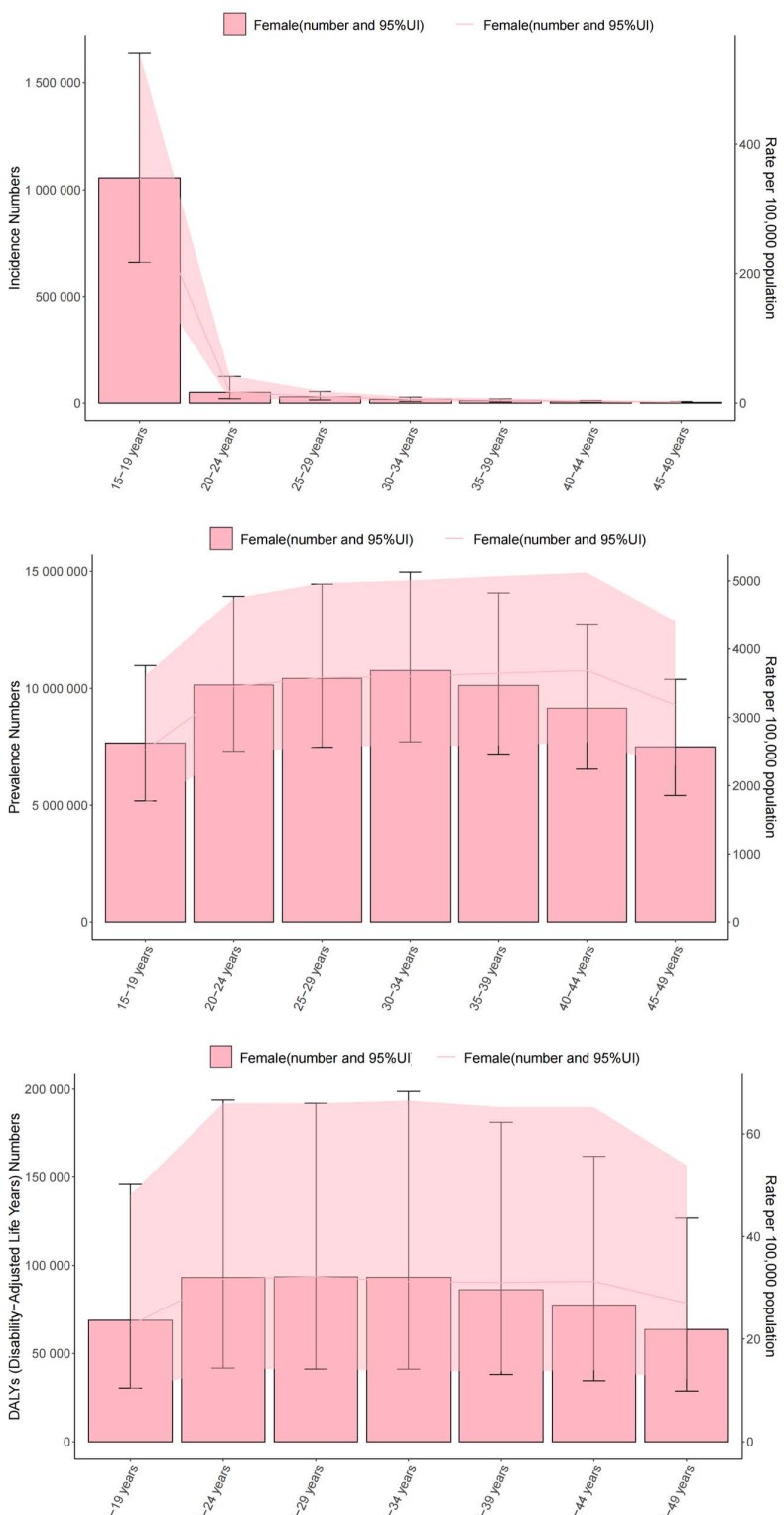

**Fig 2. Assessing the burden of polycystic ovarian syndrome among women aged 15-49 yearsin females: age-stratified incidence, prevalence, and DALYs.**

**Strength and limitations of this study**

The data for this study is derived from GBD 2021. It covers a wide range of topics, but it will inevitably be affected by reporting deviations and data quality differences, especially in low- and middle-income countries. These differences may have an impact on the accuracy of our estimation results and the universality of the research results. During the study period, the diagnosis and detection methods of polycystic ovary syndrome may have changed, and the diagnosis and recording standards of PCOS may vary in different regions. These factors may affect the comparability of epidemiological trends and the accuracy of predictions. In addition, our forecast of the burden of polycystic ovary syndrome in 2035 is based on historical trends and may not fully take into account the potential impact of changes in public health policies, technological advances, or unforeseen global events on the trajectory of the disease. These factors may significantly change future disease trends. Despite these limitations, this is the most comprehensive epidemiological report on the burden of polycystic ovary syndrome in women of childbearing age in the world and China from 1990 to 2021, and predicts the burden of disease until 2035.

**Conclusion**

This study demonstrates that the disease burden of PCOS, a major health challenge for women of reproductive age globally and in China, is showing a significant upward trend. Despite improvements in diagnostic capacity and a slight decrease in incidence in China, the increase in its prevalence and associated disease burden cannot be ignored. With the advancement of globalization and modernization, especially in high SDI countries and regions, the prevalence trend of PCOS will become more prominent, bringing a greater public health burden. Therefore, future public health strategies should strengthen early identification, health management, and lifestyle interventions for PCOS. In addition, the advancement of cross-national data sharing and multicenter studies will help us to understand the global burden of PCOS more comprehensively and provide a scientific basis for the development of effective prevention and treatment measures.

**Supporting information**

**S1 Table. Global and regional incidence of polycystic ovarian syndrome among women aged 15–49 years: a comprehensive analysis from 1990 to 2021.**
(DOCX)

**S2 Table. Global and regional prevalence of polycystic ovarian syndrome among women aged 15–49 years: a comprehensive analysis from 1990 to 2021.**
(DOCX)

**S3 Table. Global and regional DALYs of polycystic ovarian syndrome among women aged 15–49 years: a comprehensive analysis from 1990 to 2021.**
(DOCX)

**S4 Table. Global age-specific incidence of polycystic ovary syndrome in women of childbearing age: 2021 analysis.**
(DOCX)

**S5 Table. Global age-specific prevalence of polycystic ovary syndrome in women of childbearing age: 2021 analysis.**
(DOCX)

**S6 Table. Global age-specific DALYs of polycystic ovary syndrome in women of childbearing age: 2021 analysis.**
(DOCX)

**S7 Table. Global 2035 projected incidence analysis of polycystic ovary syndrome in women of reproductive age.**
(DOCX)

**S8 Table. Analysis of the Projected global overall age-standardized incidence of polycystic ovary syndrome in women of reproductive age in 2035.**
(DOCX)

**S9 Table. Global 2035 projected prevalence analysis of polycystic ovary syndrome in women of reproductive age.**
(DOCX)

**S10 Table. Analysis of the Projected global overall age-standardized Prevalence of polycystic ovary syndrome in women of reproductive age in 2035.**
(DOCX)

**S1 Fig. Projected global incidence of polycystic ovary syndrome in women of childbearing age.**
(TIF)

**S2 Fig. Projected global prevalence of polycystic ovary syndrome in women of childbearing age.**
(TIF)

**S3 Fig. Forecasting the incidence of polycystic ovary syndrome in Chinese women of childbearing age.**
(TIF)

**S4 Fig. Forecasting the prevalence of polycystic ovary syndrome in Chinese women of childbearing age.**
(TIF)

## Acknowledgments

We would also like to thank the countless individuals who have contributed to the Global Burden of Disease Study 2021 in various capacities.

## Author contributions

**Conceptualization:** Xiaodan Zhang, Junping Liu.

**Data curation:** Junping Liu.

**Funding acquisition:** Xiaodan Zhang.

**Methodology:** Xiaodan Zhang, Yan Cai.

**Resources:** Yan Cai, Jiaying Li.

**Software:** Xiaodan Zhang, Yan Cai, Jiaying Li.

**Supervision:** Junping Liu.

**Visualization:** Junping Liu.

**Writing – original draft:** Xiaodan Zhang, Junping Liu.

**Writing – review & editing:** Xiaodan Zhang, Junping Liu.

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
