## [Decision Letter · Decision Letter 0]

Dear Dr. xiaodan,

Thank you for submitting your manuscript to PLOS ONE. After careful consideration, we feel that it has merit but does not fully meet PLOS ONE’s publication criteria as it currently stands. Therefore, we invite you to submit a revised version of the manuscript that addresses the points raised during the review process.

**ACADEMIC EDITOR:**

We look forward to receiving your revised manuscript.

Kind regards,

Wan-Xi Yang, Ph.D.

Academic Editor

PLOS ONE

Journal Requirements:

2**.** Thank you for stating the following financial disclosure:

“This work was supported by the National Natural Science Foundation of China, "Construction and validation of different cardiovascular risk stratification early warning models for patients with essential hypertension based on Chinese medicine pulse detection" (NO.81973749);Construction of Ye Yumei Pudong Famous Chinese Medicine Workshop (PDZY-2021-1005);Shanghai Municipal Healthcare Commission Healthcare Program (202340155); Shanghai Pudong New District Clinical TCM Characteristic Discipline Construction Grant (YC-2023-0611); Shanghai Pudong New District Health System Leading Talent Training Program (PWRl2023-10).”

4. Please include your tables as part of your main manuscript and remove the individual files. Please note that supplementary tables (should remain/ be uploaded) as separate "supporting information" files

Reviewers' comments:

Reviewer's Responses to Questions

**Comments to the Author**

1. Is the manuscript technically sound, and do the data support the conclusions?

Reviewer #1: Yes

Reviewer #2: Yes

2. Has the statistical analysis been performed appropriately and rigorously?

Reviewer #1: I Don't Know

Reviewer #2: Yes

3. Have the authors made all data underlying the findings in their manuscript fully available?

Reviewer #1: Yes

Reviewer #2: Yes

4. Is the manuscript presented in an intelligible fashion and written in standard English?

Reviewer #1: Yes

Reviewer #2: Yes

Reviewer #1: The articles is focused on PCOS problem in Chinese popul;ation and the progression during the time. I recomended authors to discuss the problem of PCOS with exposition to EDCs and other environmental pollution and include some figures to sumarrize data.

Reviewer #2: Dear author,

Thank you for the great effort in summarising such an important data, indicating appropriately limitations and possible directions regarding findings, though with caution as it is a condition with very high heterogenicity, and thus with data not comparable many times.

About the review, I would address important issues to take into account, such as:

- Review "Results" in abstract, regarding punctuation, structure and use of incidence and prevalence.

- Line 72 - I would better define as "hyperandrogenism, oligoanovulation and polycystic ovarian morphology" (based on the latest PCOS guideline in 2023 (https://www.monash.edu/__data/assets/pdf_file/0003/3379521/Evidence-Based-Guidelines-2023.pdf)), and I would suggest to complete the information about impact and risks with the initial content of the guideline (dermatological burden, cardiometabolic, and so on)

- Line 185 and other titles: PCOS with capital letters

- Lines 189 and 190 review grammar please

- From line 200 (215, 238, ...) onwards I identify many "prevalence" words when they should indicate "incidence". I would suggest to review it carefully as it doesn't match with tables and figures.

Thank you for submitting the manuscript and I hope you find the review useful.

And thank you again for this study and conclusions. Efforts in PCOS with the most recent guidelines aim at pursue common criteria used worldwide so the data become comparable.

**Do you want your identity to be public for this peer review?** For information about this choice, including consent withdrawal, please see our Privacy Policy

Reviewer #1: No

Reviewer #2: No

---

## [Author Response · Author response to Decision Letter 1]

11 May 2025

Dear Reviewer:

Greetings! Thank you very much for your valuable comments on our paper. We have carefully revised the relevant questions, and the following are the responses to your comments:

For the first question:

We have explored in detail the relationship between PCOS and environmental endocrine disruptors (EDCs) and other environmental pollution. Specifically, we added that although the incidence of PCOS in China has decreased, its prevalence and disease burden have increased significantly by 86.95% and 86.56%, respectively. We propose that this increase may be related to the rise in EDCs.

EDCs are widely present in various products and pollutants, such as plastics, pesticides, cosmetics, and industrial pollutants, and can adversely affect the endocrine system. For instance:

Exposure to bisphenol A (BPA) can cause hyperandrogenemia and ovarian polycystic changes. The underlying mechanism involves the abnormal activation of estrogen receptor β and the dysregulation of the insulin signaling pathway.

The concentration of phthalate metabolites in the urine of PCOS patients is significantly higher than that in healthy individuals and is positively correlated with insulin resistance and oxidative stress.

In a group of women with long-term exposure to industrial pollutants, such as polychlorinated biphenyls (PCBs), the occurrence of ovarian cysts may be associated with an increase in EDCs. Additionally, in a female cohort exposed to industrial pollutants like PCBs, the risk of ovarian cysts and the levels of inflammatory factors were significantly higher.

It should also be noted that during China's rapid industrialization, the continuous increase in EDCs may partially account for the rising trend in PCOS prevalence.

For the second question:

As you suggested, we have defined PCOS as "hyperandrogenism, oligoovulation, and polycystic ovarian morphology" and supplemented this definition with the 2023 international evidence-based guidelines.

Furthermore, we have refined the information on the impacts and risks of PCOS. PCOS is the leading cause of anovulatory infertility and is associated with an increased risk of cardiovascular disease, endometrial cancer, and significant psychosocial impairment. It is also accompanied by skin manifestations such as hirsutism, acne, and alopecia.

Regarding the similarity check:

We have checked the similarity rate of the manuscript. The result of the thesis check is 13%, and the file name of the check report is "similarity rate.pdf". We have uploaded the similarity check result.

Thank you again for your interest and support of our paper, and please feel free to advise us of any further comments and suggestions.

XiaoDan ZHANG

---

## [Decision Letter · Decision Letter 1]

Dear Dr. xiaodan,

Thank you for submitting your manuscript to PLOS ONE. After careful consideration, we feel that it has merit but does not fully meet PLOS ONE’s publication criteria as it currently stands. Therefore, we invite you to submit a revised version of the manuscript that addresses the points raised during the review process.

**ACADEMIC EDITOR**
**A minor revision is needed. Please revise the manuscript accordingly.**

We look forward to receiving your revised manuscript.

Kind regards,

Wan-Xi Yang, Ph.D.

Academic Editor

PLOS ONE

Journal Requirements:

Additional Editor Comments:

Editorial comments:

As you could see, one of the reviewers had minor concerns on your revision. Please revised it accordingly.

Reviewers' comments:

Reviewer's Responses to Questions

**Comments to the Author**

Reviewer #1: All comments have been addressed

Reviewer #2: (No Response)

2. Is the manuscript technically sound, and do the data support the conclusions?

Reviewer #1: Yes

Reviewer #2: Yes

3. Has the statistical analysis been performed appropriately and rigorously?

Reviewer #1: Yes

Reviewer #2: Yes

4. Have the authors made all data underlying the findings in their manuscript fully available?

Reviewer #1: Yes

Reviewer #2: Yes

5. Is the manuscript presented in an intelligible fashion and written in standard English?

Reviewer #1: Yes

Reviewer #2: Yes

Reviewer #1: The manuscript entitled: Global and Chinese epidemiologic study of polycystic ovary syndrome in women of

childbearing age, 1990-2021, and projections to 2035: based on the Global Burden of

Disease 2021 study. Authors revision meet all my acceptation regarding the paper.

Reviewer #2: Than you very much for the corrections and the text added. However, in my previous comments I suggested a review of the use of prevalence and incidence, since both were interchanged at some points (for example, line 200, were prevalence was used twice for different purposes). Once this is corrected, I consider the article recommended for publication. Thank you very much again for the effort and very interesting study and analysis.

**Do you want your identity to be public for this peer review?** For information about this choice, including consent withdrawal, please see our Privacy Policy

Reviewer #1: No

Reviewer #2: No

---

## [Author Response · Author response to Decision Letter 2]

9 Jul 2025

Dear Editors

Thank you for reviewing our manuscript. We have addressed all reviewers' comments in the revised manuscript.

Response to Journal Requirements Regarding References:

As requested by the journal, we have thoroughly reviewed the reference list in our manuscript to verify its completeness and correctness. We specifically checked for any potential retracted papers cited within our references and can confirm that no retracted articles are cited in our manuscript. Therefore, no changes were required to the reference list in accordance with this specific point.

All cited references remain appropriate and support the claims made in the text.”

Dear Reviewer 2,

Thank you very much for your additional feedback and for highlighting the confusion in the usage of prevalence and incidence in the manuscript. We sincerely appreciate your meticulous review and apologize for the oversight in addressing this point in our previous response. Your comment is highly valuable for improving the clarity and accuracy of our work.

We have carefully reviewed the entire manuscript to identify all instances where prevalence and incidence were interchanged. As you correctly pointed out (e.g., in line 200 of the original version), prevalence was used incorrectly in place of incidence. To address this:

①We have systematically replaced "prevalence" with "incidence" in the relevant sections, including line 200, to ensure consistency and proper terminology usage.

②This correction has been highlighted in the Revised Manuscript with Track Changes, marked in red text for easy reference (specifically, in the tracked changes version). We have ensured that the distinction between prevalence (as a measure of existing cases) and incidence (as a measure of new cases) is now clear throughout the text.

With these revisions, we believe the concern you raised has been fully resolved. Your suggestion has significantly enhanced the quality of our article, and we are grateful for your expertise. Thank you once again for your time, constructive criticism, and for considering our study as "very interesting." We hope that the manuscript now meets the standards for publication and look forward to your final evaluation.

Sincerely

XiaoDan ZHANG

---

## [Editor Report · Decision Letter 2]

Global and Chinese epidemiologic study of polycystic ovary syndrome in women of childbearing age, 1990-2021, and projections to 2035: based on the Global Burden of Disease 2021 study

PONE-D-25-06617R2

Dear Dr. xiaodan,

We’re pleased to inform you that your manuscript has been judged scientifically suitable for publication and will be formally accepted for publication once it meets all outstanding technical requirements.

Kind regards,

Wan-Xi Yang, Ph.D.

Academic Editor

PLOS ONE

Additional Editor Comments (optional):

The author had revised the manuscript accordingly. No further revision is needed.
---

## [Editor Report · Acceptance letter]

PONE-D-25-06617R2

PLOS ONE

Dear Dr. xiaodan,

I'm pleased to inform you that your manuscript has been deemed suitable for publication in PLOS ONE. Congratulations! Your manuscript is now being handed over to our production team.

Kind regards,

on behalf of

Dr. Wan-Xi Yang

Academic Editor

PLOS ONE